# Report of a Novel Molecular Profile in Malignant Insulinoma

**DOI:** 10.3390/jcm12041280

**Published:** 2023-02-06

**Authors:** Laura Burns, Bita Naimi, Matthew Ronan, Huihong Xu, Horst Christian Weber

**Affiliations:** 1Department of Medicine, Boston University School of Medicine and Boston Medical Center, Boston, MA 02118, USA; 2Boston University School of Medicine, Boston, MA 02118, USA; 3VA Boston Healthcare System, Boston, MA 02130, USA; 4Department of Medicine, Harvard Medical School, Boston, MA 02115, USA

**Keywords:** hyperinsulinemic hypoglycemia, malignant insulinoma, pancreatic neuroendocrine tumor, somatostatin receptor, 68 Gallium DOTATATE PET/CT, molecular imaging, molecular profile of pNET

## Abstract

Pancreatic neuroendocrine tumors, or pNETs, represent a rare and clinically heterogenous subset of pancreatic neoplasms. One such pNET, the insulinoma, is found to be malignant in just 4% of all insulinomas. Due to the exceedingly uncommon occurrence of these tumors, there is controversy regarding the optimal evidence-based management for these patients. We therefore report on a 70-year-old male patient admitted with 3 months of episodic confusion with concurrent hypoglycemia. The patient was found to have inappropriately elevated endogenous insulin levels during these episodes, and somatostatin-receptor subtype 2 selective imaging revealed a pancreatic mass metastatic to local lymph nodes, spleen, and the liver. Fine needle aspiration of pancreatic and liver lesions confirmed the diagnosis of a low grade pancreatic neuroendocrine tumor. Molecular analysis of tumor tissue revealed a novel mutational profile consistent with pNET. The patient was initiated on octreotide therapy. However, treatment with octreotide alone demonstrated limited efficacy in controlling the patient’s symptoms, prompting consideration of other therapies.

## 1. Introduction

The population incidence of neuroendocrine tumors has been steadily rising since the 1970s as compared to the relatively stagnant population incidence of all malignant neoplasms, per the National Cancer Institute’s Surveillance, Epidemiology, and End Results (SEER) database [1]. Of this increasing incidence of neuroendocrine tumors, the majority of the increase is attributable to localized NETs, as opposed to those with regional or distant metastases [1]. Neuroendocrine tumors may arise from many different tissues in the body and represent a heterogenous group. Aside from lung NETs, the most common subtypes of NETs are gastrointestinal in nature, including neoplasms arising from the luminal portion of the gastrointestinal (GI) tract as well as pancreatic endocrine cells [1]. Pancreatic neuroendocrine tumors (pNETs) are commonly subdivided into two groups: functioning and non-functioning tumors. Functioning pNETs are those which secrete biologically active peptide hormones and present with clinically typical symptoms. One such pNET, the insulinoma, presents with a composite of classic symptoms termed Whipple’s Triad: fasting hypoglycemia (glucose < 50 mg/dL), symptoms during the hypoglycemic episodes, and relief of symptoms with the administration of glucose [2]. Insulin release is tightly regulated under normal physiologic conditions. In the case of the insulinoma, however, the release of insulin becomes uncoupled from glucose metabolism, leading to a state of hyperinsulinemic hypoglycemia [3]. Recent advances in molecular genetics and radiologic imaging modalities have permitted further characterization of these pNETs [4]. We bring forward the case of a novel mutational profile in the malignant insulinoma of a 70-year-old male which will enhance the knowledge of the molecular landscape of pNETs, specifically of insulinomas.

## 2. Case

A 70-year-old male with a history of gastroesophageal reflux disease (GERD) presented for admission at the request of his endocrinologist for evaluation of recurrent hypoglycemic episodes. The patient was in his usual state of health until approximately 3 months prior to admission, when he began to develop episodes of sweating, generalized weakness, nausea, and confusion. These episodes became progressively more frequent and severe over the next three months, at which point the patient had such a severe episode of generalized weakness that his wife brought him to the local Emergency Department (ED), where he was found to have a blood glucose of 35 mg/dL. The patient was treated with a glucose load which resolved his symptoms, and he was discharged home with a glucometer and a recommendation to carry small snacks with him in case of symptomatic hypoglycemia. The patient’s wife recorded early morning glucose levels frequently around 40 mg/dL over the following month which coincided with episodes of confusion, diaphoresis, and generalized weakness. The patient began waking for night-time snacks to mitigate these episodes. At the recommendation of their endocrinologist, the patient was admitted to our institution for further assessment of these hypoglycemic episodes.

On admission, physical examination revealed a well-developed male in no acute distress without abnormal findings. All vital signs were stable and within normal limits. Labs were notable for a blood glucose of 116 mg/dL, aspartate aminotransferase (AST) of 51 units per liter (U/L), and otherwise unremarkable laboratory studies including serum levels for metanephrines, normetanephrines, parathyroid hormone (PTH), thyroid-stimulating hormone (TSH), cortisol, and prolactin. The patient underwent a planned 72-h fast to investigate for the presence of an insulinoma, given his history of Whipple’s Triad. Approximately 12 h after the fast began, the blood glucose level declined to 36 mg/dL and the patient experienced shakiness and demonstrated confusion, at which time relevant blood samples were obtained, with results displayed in Table 1.

Recurrent hyperinsulinemic hypoglycemia has a broad range of possible differential diagnoses in the pediatric and adult patient population [5]. For instance, it may also be seen in autoimmune-mediated insulin release [6]; however, insulin autoantibody titers were negative in this patient. Surreptitious endogenous insulin administration was also considered, but C-peptide would have been discordantly lower than our measured level. A negative sulfonylurea urine panel also ruled out oral insulinogenic agents as the culprit. This patient’s proinsulin level was significantly higher than the upper limit of normal, particularly compared to the elevated insulin level. Markedly elevated proinsulin and an increased ratio of proinsulin-to-insulin has been associated with malignant insulinoma as opposed to benign insulinoma [7].

We next pursued cross-sectional imaging to help us distinguish between diffuse islet cell hyperplasia, which can cause a noninsulinoma pancreatogenous hypoglycemia syndrome (NIPHS), from an insulinoma [8]. Magnetic resonance imaging (MRI) of the abdomen (Figure 1A) revealed multiple liver masses, the largest of which was 7.6 by 5.2 cm (cm) in the right lobe, in addition to a 5.3 by 4.3 cm mass in the spleen, three masses in the pancreatic tail, and regional lymphadenopathy, with imaging characteristics typical for neuroendocrine tumors. These findings were highly concerning for the diagnosis of malignant insulinoma in the clinical context of hyperinsulinemic hypoglycemia.

To assess for other metastatic lesions of neuroendocrine origin, the patient then underwent a 68-Gallium DOTATATE positron emission tomography/computed tomography (PET/CT) scan, which uses a radiopharmaceutical tracer that binds selectively to the somatostatin-receptor subtype 2 which is frequently expressed by pNETs [9]. The results, pictured in Figure 1B, showed highly increased uptake in the pancreas, multiple liver lesions, spleen, and intra-abdominal lymph nodes, which suggested in this clinical context that the masses identified on this molecular abdominal imaging likely represent metastatic malignancy of pancreatic origin.

The Gastroenterology service was consulted and an endoscopic ultrasound with fine-needle aspiration (FNA) of both a liver and pancreatic lesion was performed.

Smears and cell blocks were prepared from the fine needle aspiration sampling of both liver and pancreatic lesions. Papanicolaou (Pap) stained smears, as well as Hematoxylin and Eosin (H&E) stained cell block tissue sections were used for further examination. Cells from either liver or pancreas lesions share a similar cytomorphology and immunocytochemical profile. These single or loosely grouped lesional cells are relatively small and uniform in size with eccentrically located round or oval nuclei with finely stippled chromatin and moderate eosinophilic cytoplasm. Immunohistochemistry (IHC) studies revealed cells positive for pan-cytokeratin AE1/3 and neuroendocrine markers including CD56, synaptophysin, and chromogranin. However, cells were negative for beta-catenin, S100, and TTF-1. Proliferation rate as indicated by Ki67 was less than 3%. Mitotic count was less than 2 per 10 high power field. These findings suggested a well differentiated, low grade (G1) pancreatic neuroendocrine tumor (pNET) most consistent with malignant insulinoma. Selected cytopathology and IHC examinations are shown in Figure 2.

To further characterize this patient’s tumor, genomic sequencing was performed using FoundationOne^®^CDx (F1CDx; Appendix A), which is summarized in Table 2.

Initial therapy of hypoglycemic episodes included subcutaneous injections of octreotide (150 micrograms subcutaneously three times daily), a somatostatin analogue that suppresses insulin, growth hormone, and glucagon secretion. However, the monotherapy with octreotide failed to prevent the onset of hypoglycemia overnight while fasting in the few days following its initiation. The patient experienced ongoing labile blood glucose levels despite escalating doses of octreotide and the addition of diazoxide 50 mg twice daily as an adjunctive agent. The patient finally attained relief from overnight hypoglycemia (maintenance of blood glucose greater than 70 mg/dL for 8 h overnight) with a combination of octreotide 300 micrograms (mcg) subcutaneous injections three times daily, Diazoxide 100 mg twice daily, and ingestion of a slurry of cornstarch and cold water prior to bedtime. On discharge, octreotide was transitioned from three times daily dosing to a monthly depot injection of 20 mg. The patient’s treatment course was complicated by the development of lower extremity edema with the addition of diazoxide which was adequately managed with furosemide 40 mg daily. The patient was discharged in stable condition with a glucometer and outpatient follow up.

## 3. Discussion

The patient in our case was one of those approximately 1 to 3 per 10 million individuals found to have malignant insulinoma. The National Comprehensive Cancer Network’s 2020 guidelines recommend consideration of lanreotide or octreotide, somatostatin analogues, as treatment options for non-resectable, symptomatic, metastatic neuroendocrine tumors of the pancreas [10]. This guidance is based on the findings in the CLARINET and PROMID trials, both of which demonstrated improved progression-free survival with octreotide vs. placebo in midgut neuroendocrine tumors [11,12]. Considering this backdrop of compelling data for the use of octreotide in patients with tumors with robust concentration of SSR2 receptors, as identified in our patient, octreotide was initiated in this case. Unfortunately, the patient continued to struggle with hypoglycemic episodes despite octreotide dose escalation, which prompts consideration of other available treatment modalities. Given somatostatin analogues may also suppress secretion of glucagon, there is inconsistency in their effects on blood glucose levels. The North American Neuroendocrine Tumor Society (NANETS) 2020 Consensus Guidelines support ^177^Lu-DOTATATE Peptide Receptor Radionuclide Therapy (PRRT) for progressing pNET patients [13]. PRRT therapy utilizes Tyr3-octreotate (TATE) to bind to the SSR2 receptor and thereby deliver the attached radionuclide directly to the malignant cells. Small case series have shown this treatment to be effective in controlling hypoglycemic symptoms in patients with malignant insulinoma [14]. If our patient is deemed to progress in the outpatient setting clinically or radiographically, this data would support trialing PRRT therapy, in alignment with 2020 NCCN guidelines. In addition, everolimus, an mTOR inhibitor, has been reported effective in controlling insulin-induced hypoglycemia in malignant insulinomas refractory to somatostatin analogues, and could be considered in this patient [15,16].

Given the rarity of insulinoma, and particularly metastatic insulinoma, less is known about its genomic profile. In a study using whole-exome gene sequencing to compare the genomic profiles of 84 insulinomas with 127 non-functioning pNETs (NF-pNETs), *Yin Yang 1* (*YY1*) mutations were unique to a minority of insulinomas (25%) and not noted in any NF-pNETs. *YY1* is a transcription factor involved in various functions including cell growth and development. Mutation of the third zinc finger region of *YY1* has been shown to alter its transcriptional activity, which promotes tumorigenesis in insulinoma [17]. Unfortunately, *YY1* testing is not yet included in the comprehensive molecular panel for which this patient’s tumor tissue was submitted (Appendix A). In contrast, the NF-pNET group contained frequent mutations in *MEN1* (42%)*, DAXX* (21%)*, ATRX* (13%)*,* and mTOR pathway genes (14%) [18]. Literature becomes even more sparse when narrowing focus to specifically malignant insulinomas. One group examined a cohort of 35 primary insulinomas which included five metastatic insulinomas. Of these, all five retained *ATRX* expression, while one had *DAXX* loss [19]. It was posited from these results that *ATRX* and *DAXX* mutations, which are typically more characteristic of NF-pNETs, may be more common in malignant insulinomas than in their less aggressive counterparts [19]. Groups sequencing larger cohorts including insulinomas have reported *ATRX* mutations in insulinoma very rarely [20]. Affirming these reports, the L2027R *ATRX* mutation noted in our patient affords additional credence to this suggestion that chromosomal instability from *DAXX*/*ATRX* mutations may represent an important genomic basis for the aggressive behavior of malignant insulinoma [19].

This patient’s tumor is the first malignant insulinoma to our knowledge to possess simultaneous mutations in *ATRX*, *ROS1*, and *KMT2A*. For other malignancies harboring these genomic alterations, additional investigation and trials of targeted therapeutics have yielded promising results. Innovation in novel therapeutics for insulinoma, particularly for the rare and high-mortality malignant insulinoma, will hinge on gathering and reporting molecular profiles such as this one (Appendix A).

## 4. Conclusions

The diagnosis of a pancreatic neuroendocrine tumor requires a methodical approach with a keen focus on ruling out more common disease processes, particularly in the case of insulinoma. Given the rarity of insulinoma and the relatively recent advent of comprehensive molecular testing, there are few molecular profiles of insulinoma in the literature. We therefore present a novel mutational profile for consideration. Additional molecular data such as that reported here will be critical to achieving therapeutic advances for malignant insulinoma.

## Figures and Tables

**Figure 1 jcm-12-01280-f001:**
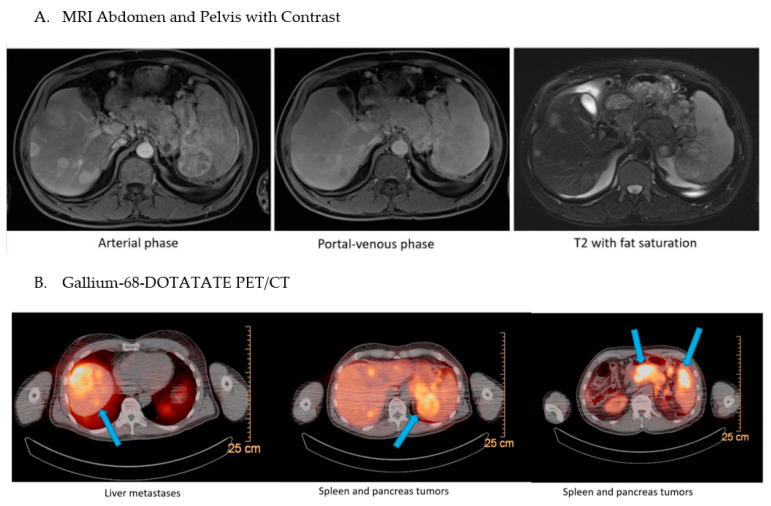
Imaging Studies of the Abdomen and Pelvis. (**A**) For diagnostic purposes, MRI was obtained using gadolinium contrast. Representative images of three phases are shown as indicated. (**B**) The patient was subjected to a Gallium-68-DOTATATE PET/CT and several areas of high-intensity uptake are delineated by arrows in the series of cross-sections.

**Figure 2 jcm-12-01280-f002:**
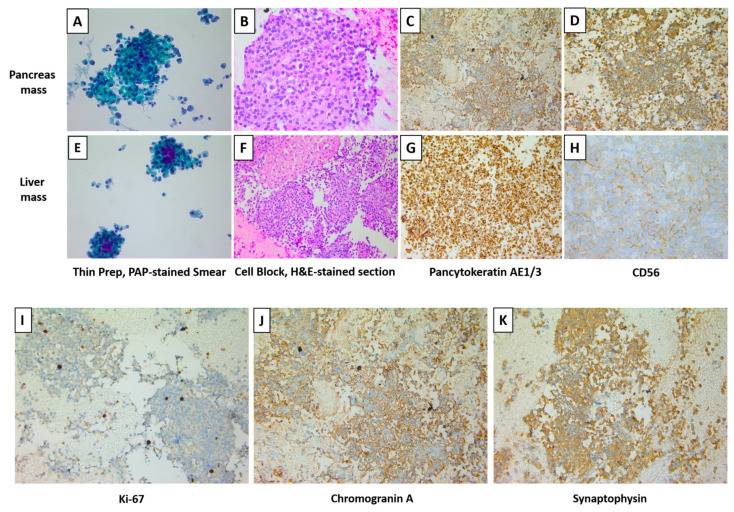
The Cytomorphology, Immunocytochemical, and Biomarker profile of both Liver and Pancreas Lesions. Paraffin embedded cellblock material from fine needle aspiration of the pancreatic and liver masses. (**A**,**E**): Cellular aspirate of loosely cohesive clusters of cells and single cells (Papanicolaou stain). (**B**,**F**): Cell block tissue sections (H&E stain). (**C**,**G**): The cells are positive for pan-cytokeratin stain. (**D**,**H**): The cells are positive for neuroendocrine marker CD56. (**I**–**K**): Paraffin embedded cellblock material from fine needle aspiration of the pancreatic mass was subjected to immunohistochemistry staining for Ki-67, chromogranin A, and synaptophysin, all of which are positive. The original magnifications are as follows: ×400 (**A**), ×400 (**B**), ×200 (**C**), ×400 (**D**), ×400 (**E**), ×200 (**F**), ×200 (**G**), ×400 (**H**), ×400 (**I**), ×200 (**J**), and ×200 (**K**).

**Table 1 jcm-12-01280-t001:** Laboratory Evaluation at 12 Hours Fasted.

Laboratory Test	Patient’s Result	Reference Range
Glucose	36 mg/dL	70–100 mg/dL
Glucagon	265 pg/mL	50–100 pg/mL
Insulin	21.2 mIU/mL	2–20 mIU/mL
Proinsulin	912 pmol/L	6.5–8.9 pmol/L
C-peptide	5.2 mg/mL	0.5–2 mg/mL
Insulin Auto Antibody Titer	<0.4 nU/mL	<0.4 nU/mL
Insulin-like Growth Factor 2 (IGF-2)	304 ng/mL	267–616 ng/mL
Sulfonylurea Screen	Negative	Negative

The patient underwent a monitored fast and when hypoglycemia was noted at 12 h into the fasting test, a set of labs was collected which demonstrated concomitantly elevated levels of the biomarkers proinsulin, insulin, and C-peptide, suggestive of a diagnosis of insulinoma.

**Table 2 jcm-12-01280-t002:** Molecular Profiling of Liver Metastasis.

Gene Name	Protein	Cellular Function	Mutation in This Patient ^a^
*ATRX*	Alpha-thalassemia-mental retardation, X-linked	Maintains the integrity of heterochromatin	Yes L2027R
*DAXX*	Death domain associated protein	Acts as a histone chaperone	None
*MEN1*	Multiple endocrine neoplasia I	Participates in transcriptional regulation	None
*mTOR* pathway genes (*DEPDC5*, *mTOR*, *PTEN*, *TSC1*, *TSC2*)	Mammalian target of rapamycin	Growth regulatory pathway	No
*HGF*	Hepatocyte growth factor	Stimulates mitogenic activity	Amplification (equivocal)
*EPHB4*	Ephrin type-B receptor 4	Stimulates vascular development	Yes A246S
*ROS1*	Proto-oncogene tyrosine-protein kinase ROS	Involved in cell growth/proliferation via unclear mechanism	Yes R2096W
*KMT2A* (MLL)	Lysine methyltransferase 2A	Transcriptional regulator	Yes R12G

Liver metastatic tissue was obtained by fine needle aspiration using endoscopic ultrasonography as described and sent for molecular testing to FoundationOne^®^CDx (F1CDx ^a^; Appendix A).

## Data Availability

Not applicable.

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
