# Peer review of "Report of a Novel Molecular Profile in Malignant Insulinoma"

_jcm, 2023, doi:10.3390/jcm12041280_

Round 1

Reviewer 1 Report

The case is really interesting and is well presented, however lacks information on the point which is deemed most important at the end of the introduction: "Enhance the knowledge of the molecular landscape of insulinomas". No sequencing data is provided in the case report, but this data is referred to in the discussion. It seems part of the case report is missing? It seems the discussion is actually a separate review. Additional case information is scattered through the review section, including sequencing data, which should be described in the case.  

It is not clear to this reviewer why this case report is combined with a review that is not completely about (malignant) insulinoma, but also discusses pNEN and other NENs, and even makes a very lengthy comparison of pancreatic NET with PDAC (which is not relevant). In this case, the diagnosis was easily made and no diagnostic uncertainty was present, so why is it necessary to review all different diagnostic modalities?

Also, most indolent insulinomas have a different mutational profile than pancreatic NETs (it seems table 2 compares with pancreatic NETs). However, later in the discussion, the key study by Hong et al. is mentioned (YY1 mutations).

This reviewer would suggest to present this text as two separate manuscripts.

- One focusing on this very interesting case, starting with a brief introduction on the epidemiology and background of malignant insulinoma, followed by the case report, and perhaps concluding with a brief overview of the relevant literature and the lessons learned from this case. I do think the case is really interesting and worth getting out there.

- Another manuscript could be made out of the discussion, which could be about the diagnosis and treatment of pNET, including differential diagnoses and diagnostic pitfalls. However, this needs extensive rewriting.

Other points

- 5 figures is too many for a case report. I would suggest to combine all radiology and histology into 2 main figures.

- The term neuroendocrine neoplasm suggests you are also talking about neuroendocrine carcinoma, which is a completely different tumor type. "NET" refers to only the tumors.

- Line 83 concercing = confirming?

- Line 134-137. To assess ... to the somatostatin-receptor subtype 2 present in almost all low 136 grade pNEN cells except non-malignant insulinomas. This is not correct, see for example PMID: 26923247

- Line 140, 68-Gallium DOTATATE PET/CT positivity is not specific for malignant insulinoma.

- line 200 patients with widely metastasized disease might not be fit or considered for surgery, which can bias these results

Author Response

Reviewer 1:

The case is really interesting and is well presented, however lacks information on the point which is deemed most important at the end of the introduction: "Enhance the knowledge of the molecular landscape of insulinomas". No sequencing data is provided in the case report, but this data is referred to in the discussion. It seems part of the case report is missing? It seems the discussion is actually a separate review. Additional case information is scattered through the review section, including sequencing data, which should be described in the case.  

It is not clear to this reviewer why this case report is combined with a review that is not completely about (malignant) insulinoma, but also discusses pNEN and other NENs, and even makes a very lengthy comparison of pancreatic NET with PDAC (which is not relevant). In this case, the diagnosis was easily made and no diagnostic uncertainty was present, so why is it necessary to review all different diagnostic modalities?

Also, most indolent insulinomas have a different mutational profile than pancreatic NETs (it seems table 2 compares with pancreatic NETs). However, later in the discussion, the key study by Hong et al. is mentioned (YY1 mutations).

This reviewer would suggest to present this text as two separate manuscripts.

- One focusing on this very interesting case, starting with a brief introduction on the epidemiology and background of malignant insulinoma, followed by the case report, and perhaps concluding with a brief overview of the relevant literature and the lessons learned from this case. I do think the case is really interesting and worth getting out there.

- Another manuscript could be made out of the discussion, which could be about the diagnosis and treatment of pNET, including differential diagnoses and diagnostic pitfalls. However, this needs extensive rewriting.

We appreciate very much the thoughtful recommendations put forward by Reviewer 1 and we have therefore significantly modified this manuscript accordingly. Primarily, we have restructured this manuscript to focus on the case of the novel molecular profile and have removed the separate review portion from the manuscript for improved clarity and organization. We have moved a discussion of the molecular profile of this malignant insulinoma into the case itself. We feel these changes have resulted in greatly improved clarity and organization of the manuscript as Reviewer 1 suggested. We are pleased to return a significantly revised manuscript which highlights the salient findings of this case, which we appreciate also interested Reviewer 1.

Other points

- 5 figures is too many for a case report. I would suggest to combine all radiology and histology into 2 main figures.

We agree with Reviewer 1 and all radiology and histology images have now been combined into two figures as recommended (radiology in Figure 1, cytopathology and IHC in Figure 2). The MIP image of the PET/CT scan and EUS images were removed.

- The term neuroendocrine neoplasm suggests you are also talking about neuroendocrine carcinoma, which is a completely different tumor type. "NET" refers to only the tumors.

We agree with Reviewers 1’s comment. Accordingly, the terminology has been updated for precision, and now “NET” is used appropriately in lieu of neoplasm.

- Line 83 concercing = confirming?

We have removed the entire sentence as it did not improve clarity of the paragraph.

- Line 134-137. To assess ... to the somatostatin-receptor subtype 2 present in almost all low 136 grade pNEN cells except non-malignant insulinomas. This is not correct, see for example PMID: 26923247

- Line 140, 68-Gallium DOTATATE PET/CT positivity is not specific for malignant insulinoma.

Reviewer 1 is correct in stating that positive 68-Gallium DOTATATE PET/CT is not specific for malignant insulinoma.

Therefore, we have corrected these statements and amended the paragraph to read “which uses a radiopharmaceutical tracer that binds selectively to the somatostatin-receptor subtype 2 which is frequently expressed by pNETs” and “which suggested in this clinical context that the masses identified on this molecular abdominal imaging likely represent metastatic malignancy of pancreatic origin.”

- line 200 patients with widely metastasized disease might not be fit or considered for surgery, which can bias these results

Thank you for this insightful comment. The paragraph containing this line was removed as part of the restructuring of the manuscript involving removal of the review portion.

Reviewer 2 Report

The paper of Laura Burns and colleague report a molecular profile in one stage IV malignant insulinoma patient.

Some objection to acceptance

- some statements are inaccurate

--NENs not arise mostly in the luminal gastrointestinal tract; neuroendocrine cells are widely scattered in all body tissue in the luminal site or into the wall)

-- acronym could be better reported when used

--octreotide isn't the first line therapy for insulinoma but could be used in some cases

-- the use of reference n. 3 in the introduction is not correct.  The paper mostly review NON oncological hypoglycemia. The paper reported in oncological patients (other than mutation in MEN1) somatic mutation in T372R mutation YY1 in 10 sporadic insulinoma patients and occurrence of HKI defects in insulin neuroregulation.  

- the part of diagnostic examination and radiological picture is not necessary for the aim of the paper and doesn't add any new information.  The same for differential diagnosis because we have pathological data from primary tumors and hepatic lesion.

- no consideration is available on minimal increase in insulin value but very very high pro insulin value during hypoglycemic episode   

- discussion on the somatostatin analogues therapy in pNET need to be brought back to only insulinoma out of pNET. In particular in this case an high level of glucagon, is probably the most important aid to maintain normal glycemic level and as reported somatostatin analogues could worst the glycometabolic balance.  

- everolimus treatment isn't considered for this patient when many paper also in important medicine journal reported good results in advanced insulinoma

None of the mutation reported could change the therapeutical approach in this case.  

Author Response

Reviewer 2

We thank reviewer 2 for the insightful comments! We addressed the concerns point by point and present now our significantly revised and improved manuscript regarding the novel molecular profile in a malignant insulinoma.

--NENs not arise mostly in the luminal gastrointestinal tract; neuroendocrine cells are widely scattered in all body tissue in the luminal site or into the wall)

The reviewer 2 rightly points to a broad presence of neuroendocrine cells in a large number of tissues. This statement has been amended to correctly reflect the epidemiology of neuroendocrine tumors as described by Reviewer 2. The statement now reads “Neuroendocrine tumors may arise from many different tissues in the body and represent a heterogenous group. Aside from lung NETs, the most common subtypes of NETs are gastrointestinal in nature, including neoplasms arising from the luminal portion of the GI tract as well as pancreatic endocrine cells”

-- acronym could be better reported when used

Long-form of acronyms at first-time use have been added.

--octreotide isn't the first line therapy for insulinoma but could be used in some cases

Thank you for this significant comment! We appreciate this excellent point and have edited our statement to read: “The National Comprehensive Cancer Network’s 2020 guidelines recommend consideration of lanreotide or octreotide, somatostatin analogues, as treatment options for non-resectable, symptomatic, metastatic neuroendocrine tumors of the pancreas”

-- the use of reference n. 3 in the introduction is not correct.  The paper mostly review NON oncological hypoglycemia. The paper reported in oncological patients (other than mutation in MEN1) somatic mutation in T372R mutation YY1 in 10 sporadic insulinoma patients and occurrence of HKI defects in insulin neuroregulation.  

Thank you for this astute observation. This statement in the Introduction is now supported by an appropriate citation (Haris et al.).

- the part of diagnostic examination and radiological picture is not necessary for the aim of the paper and doesn't add any new information.  The same for differential diagnosis because we have pathological data from primary tumors and hepatic lesion.

We appreciate this comment: Accordingly, we have significantly edited and reorganized the diagnostic examination and radiological images to improve this portion of the manuscript. For instance, all radiology and histology images have now been combined into two figures and the MIP image of the PET/CT scan has been removed.

- no consideration is available on minimal increase in insulin value but very very high pro insulin value during hypoglycemic episode   

Based on the reviewer’s comment, a statement and reference has been added which note that disproportionate increase in proinsulin compared to insulin is associated with malignant insulinoma (as opposed to benign insulinoma). Lines 83-86:” This patient’s proinsulin level was significantly higher than the upper limit of normal, particularly compared to the elevated insulin level. Markedly elevated proinsulin and increased ratio of proinsulin-to-insulin has been associated with malignant insulinoma as opposed to benign insulinoma [7]. “

- discussion on the somatostatin analogues therapy in pNET need to be brought back to only insulinoma out of pNET. In particular in this case an high level of glucagon, is probably the most important aid to maintain normal glycemic level and as reported somatostatin analogues could worst the glycometabolic balance.  

This reviewer’s comment is valuable and much appreciated! We have narrowed the discussion of somatostatin analogues and acknowledged that our patient may not have responded to a somatostatin analogue due to concurrent effects on glucagon secretion. Line 176-178: “Given somatostatin analogues may also suppress secretion of glucagon, there is inconsistency in their effects on blood glucose levels. [16 17]”

- everolimus treatment isn't considered for this patient when many paper also in important medicine journal reported good results in advanced insulinoma

Thank you for pointing to this option. Consideration of everolimus is now discussed, and references to success in malignant insulinoma are provided. Line 193-195: “In addition, everolimus, an mTOR inhibitor, has been reported effective in controlling insulin-induced hypoglycemia in malignant insulinomas refractory to somatostatin analogues, and could be considered in this patient”

None of the mutation reported could change the therapeutical approach in this case.  

While none of the mutations reported in this case are presently actionable, there are so few cases of insulinoma, specifically including molecular profiles, that we feel adding this to the sparse existing body of literature may facilitate additional investigation that may benefit similar patients such as this in the future. 

Round 2

Reviewer 1 Report

Greatly improved manuscript. 

Minor points:

- Please upload higher quality figures in formatting and DPI, the text boxes seem to have different fonts/sizes, the CD56 is not interpretable. Moreover, information on magnification/size bar is missing

- This reviewer would advise to remove most text on "pNET" in general from the discussion, and focus on insulinoma. The first lines (159-161) repeat information from the introduction. 

- Although you are first to specifically report an ATRX mutation as main finding in results for malignant insulinoma, others have reported ATRX mutations in malignant insulinoma before in larger sequenced cohorts, for example see the discussion in Di Domenico et al. 2020 (PMID 33288854)

Author Response

Greatly improved manuscript. 

We thank Reviewer 1 for the very helpful recommendations on the prior version of our manuscript and are pleased to offer an additionally refined version of the manuscript with the guidance of Reviewer 1’s expertise. 

Minor points:

- Please upload higher quality figures in formatting and DPI, the text boxes seem to have different fonts/sizes, the CD56 is not interpretable. Moreover, information on magnification/size bar is missing

We have replaced the CD56 image (Figure 2 top panel, plate H) with a different image and used improved, higher resolution images for this figure (top and bottom panel).  All text boxes have been resized as needed to identical (font) sizes and boxes.

The figure legend has been modified to indicate the specific magnifications of all images used in figure 2.

“The original magnifications are as follows: x400 [A], x400 [B], x200 [C], x400 [D], x400 [E], x200 [F], x200 [G], x400 [H], x400 [I], x200 [J], and x200 [K].”

Size bars are not available in the Department of Pathology photographic equipment and could not be used.

- This reviewer would advise to remove most text on "pNET" in general from the discussion, and focus on insulinoma. The first lines (159-161) repeat information from the introduction. 

We appreciate this helpful feedback. The redundant lines 159-161 have been removed. The body of text referencing pNETs in the discussion has been reduced in order to maintain focus on insulinoma.

- Although you are first to specifically report an ATRX mutation as main finding in results for malignant insulinoma, others have reported ATRX mutations in malignant insulinoma before in larger sequenced cohorts, for example see the discussion in Di Domenico et al. 2020 (PMID 33288854)

Thank you for this helpful reference, which we have subsequently incorporated into the manuscript in line 202. We have reframed our statement to acknowledge this other reports of ATRX mutations in malignant insulinoma.
